# Signal Sequence-Dependent Orientation of Signal Peptide Fragments to Exosomes

**DOI:** 10.3390/ijms23063137

**Published:** 2022-03-15

**Authors:** Kenji Ono, Mikio Niwa, Hiromi Suzuki, Nahoko Bailey Kobayashi, Tetsuhiko Yoshida, Makoto Sawada

**Affiliations:** 1Department of Brain Function, Division of Stress Adaptation and Protection, Research Institute of Environmental Medicine, Nagoya University, Nagoya 464-8601, Aichi, Japan; hiromi_s@riem.nagoya-u.ac.jp (H.S.); msawada@riem.nagoya-u.ac.jp (M.S.); 2Department of Molecular Pharmacokinetics, Nagoya University Graduate School of Medicine, Nagoya 464-8601, Aichi, Japan; 3Institute for Advanced Sciences, Toagosei Co., Ltd., Tsukuba 300-2611, Ibaraki, Japan; mikio_niwa@mail.toagosei.co.jp (M.N.); nahoko_kobayashi@mail.toagosei.co.jp (N.B.K.); tetsuhiko_yoshida@mail.toagosei.co.jp (T.Y.)

**Keywords:** signal peptide, exosomes, intercellular communication, calmodulin

## Abstract

Signal peptides (SPs) not only mediate targeting to the endoplasmic reticulum (ER) but also play important roles as biomarkers and substances with physiological activity in extracellular fluids including blood. SPs are thought to be degraded intracellularly, making it unclear how they are transported from the ER to the extracellular fluid. In a recent study, we showed that a C-terminal fragment of the SP of a type I membrane protein, amyloid precursor protein (APP), was secreted into the extracellular fluid via exosomes using transformed HEK293 cells expressing APP SP flanking a reporter protein. In the present study, we demonstrate that a N-terminal fragment of the SP from a type II membrane protein, human placental secreted alkaline phosphatase (SEAP), is contained in exosomes and secreted into the extracellular fluid using HEK-Blue hTLR3 cells, which express both a human toll-like receptor 3 gene and an inducible SEAP reporter gene. When HEK-Blue hTLR3 cells were stimulated with a TLR3 ligand, a N-terminal fragment of SEAP SP in exosomes was increased in parallel with SEAP secretion in a concentration-dependent manner. These results indicated that SP fragments are exosomal components. In addition, migrating SP fragments were determined by characteristics of the signal–anchor sequence of membrane proteins. Furthermore, we found that SP fragments could bind to calmodulin (CALM), which is a cytosolic protein and also a component of exosomes, suggesting its involvement in the transportation of SP fragments from the endoplasmic reticulum to exosomes.

## 1. Introduction

Recent studies have demonstrated the novel potential roles of signal peptides (SPs) and their fragments as biomarkers and substances with physiological activity [1,2,3,4,5,6,7,8,9]. SP fragments derived from A-type natriuretic peptide (ANP), B-type natriuretic peptide (BNP), or C-type natriuretic peptide (CNP) are present in human circulation [1,2,3]. In healthy individuals, the plasma concentration of each SP does not correlate with its pro-peptide sibling, and responses vary during the course of ST-elevation myocardial infarction. All three SPs show evidence of being secreted from the heart, raising the possibility that measuring plasma SPs may be useful in cardiovascular disease. In addition, the SP of eosinophil cationic protein plays a role in cytotoxicity to inhibit the growth of microbials [4] and regulates the expression of mammalian genes encoding tumor growth factor-α, epidermal growth factor receptor and proinflammatory molecules [5,6]. Protease-activated receptor 1 possesses a functional and cleavable SP that plays a role in cardio-protection after ischemia and reperfusion injury [7,8]. The SP of GluK1, which is a type of kainate-type glutamate receptor and plays an important role in excitatory synaptic transmission and synaptic plasticity, directly interacts with the amino-terminal domain to inhibit the synaptic and surface expression of GluK1, indicating that the cleaved SP behaves as a ligand of GluK1 to repress forward trafficking of the receptor [9]. The functions of these SPs are unanticipated considering the classical roles of SPs.

Usually, SPs are located at the N-terminal of newly synthesizing proteins and mediate targeting to the endoplasmic reticulum (ER) membrane [10] and the insertion into the translocon for transport through the lipid bilayer [11]. SPs are cleaved from synthesizing proteins by signal peptidases [12] on the ER membrane and further cleaved by signal peptide peptidases (SPP) into two peptide fragments [13]. Although it was thought that cleaved SPs are degraded intracellularly [14], it has become clear that some SPs and their fragments have diverse functions for intra- and inter-cellular communications [15]. However, it is unclear how they are transported from the ER without being degraded. In a recent study, we showed that an SP fragment of amyloid precursor protein (APP) is secreted into the extracellular fluid via exosomes using T-REx AspALP cells, which are transformed HEK293 cells expressing APP SP flanking mature human placental secreted alkaline phosphatase (SEAP), a type II membrane protein [16]. Although the C-terminal fragment but not an N-terminal fragment of APP SP was found in exosomes, it has remained unclear whether the N-terminal fragment is contained in exosomes.

In this study, we demonstrate that the N-terminal but not C-terminal fragment of SEAP SP is contained in exosomes and secreted into the extracellular fluid using HEK-Blue hTLR3 cells that express both human toll-like receptor 3 gene and inducible SEAP reporter genes. Distinct SP sequences resulted in orientational differences in the fragment contained in the exosomes. In addition, we found that SP fragments could bind to calmodulin (CALM), which is a cytosolic protein and a component of exosomes, suggesting that CALM mediates the transportation of SP fragments from the ER to exosomes.

## 2. Results

### 2.1. Cell Systems

In a previous study, we found the C-terminal fragment but not N-terminal fragment of APP SP in exosomes was released from T-REx AspALP cells [16]. T-REx AspALP cells are Tet-repressor-expressing HEK-293 cells and express the APP SP-SEAP hybrid protein in the presence of Dox, which replaces the SP sequence of SEAP with that of APP. On the other hand, HEK-Blue hTLR3 cells are HEK293 cells co-transfected with hTLR3 gene and inducible SEAP reporter gene. HEK-Blue hTLR3 cells express the SEAP SP-SEAP hybrid protein in the presence of TLR3 ligands such as poly(I:C). In this study, we examined SP fragments in exosomes from T-REx AspALP and HEK-Blue hTLR3 cells in order to determine whether characteristics of the signal-anchor sequence of membrane proteins affect migrating SP fragment(s) using type I (APP) and type II (SEAP) membrane proteins.

### 2.2. Secretion of SEAP from HEK-Blue hTLR3 Cells after Poly(I:C) Treatment

To confirm the expression of transfected gene products including SPs in both T-REx AspALP and HEK-blue hTLR3 cells by treatment with Dox and poly(I:C), the activity of SEAP, a mature product protein derived from the transfected gene, was measured with conditioned medium (Figure 1). Since enzyme activity in conditioned medium is detected when the introduced gene product is synthesized and the SP sequence is removed and secreted, it can be used as an index for SP production. Consistent with a previous study [16], SEAP activity from T-REx AspALP cells but not T-REx Mock cells was found by treatment with Dox (Figure 1A). In addition, there was no alteration in SEAP activity from T-REx Mock and T-REx AspALP cells by treatment with poly(I:C). SEAP activity from HEK-Blue hTLR3 cells was increased by poly(I:C) treatment (Figure 1B), although it was weaker than the activity from Dox-treated T-REx AspALP cells. When HEK-Blue hTLR3 cells were treated at concentrations of 0–10 μg/mL, the SEAP activity was concentration-dependent.

### 2.3. Detection of SEAP SP in Exosomes from HEK-Blue hTLR3 Cells

In a previous study, we reported that exosomes from Dox-treated T-REx AspALP cells, which expressed APP SP as a product of the transfected gene, contained a C-terminal fragment of APP SP [16]. To confirm that exosomes from poly(I:C)-treated HEK-Blue hTLR3 cells, which expressed SEAP SP but not APP SP as a product of the transfected gene, we examined APP SP in exosomes from HEK-Blue hTLR3 cells (Figure 1C). We found the peak size at *m*/*z* 861 as the C-terminal fragment of APP SP in exosomes from Dox-treated T-REx AspALP cells but not in exosomes from poly(I:C)-treated HEK-Blue hTLR3 cells. To clarify whether exosomes from HEK-Blue hTLR3 cells contained SEAP SP, we examined SEAP SP in the exosomes by MALDI-TOF MS. We found the peak size at *m*/*z* 1277, which was increased by poly(I:C) treatment and different from the peak size for full length SEAP SP (MLLLLLLLGLRLQLSLG, *m*/*z* 1878) in the exosome fraction (Figure 2A). MS/MS analysis identified the peak at *m*/*z* 1277 +/− 6 as the N-terminal fragment of SEAP SP (MLLLLLLLGLR, MW 1267), which contains an MLL-amino acid sequence (Figure 2B). The intensity of the peak size at *m*/*z* 1277 was increased as the concentration of poly(I:C) increased (Figure 2C). These observations indicated that different SP sequences changed the orientation of the SP fragment in the exosomes.

### 2.4. Alteration in Properties of HEK-Blue hTLR3 Cells after Poly(I:C) Treatment

We examined the properties of HEK-Blue hTLR3, T-REx Mock and T-REx AspALP cells after treatment with poly(I:C); a hTLR3 ligand but found no morphological changes (Figure 3A). Additionally, the number of cells slightly decreased as the concentration of poly(I:C) increased (Figure 3B), but there was no alteration in cell viability (Figure 3C).

### 2.5. Properties of Exosomes from HEK-Blue hTLR3 Cells

To examine the properties of exosomes released from poly(I:C)-treated cells, the average size and number of released exosomes were determined by the NTA method. There were no differences in the average size with and without poly(I:C) treatment (Figure 4A). The number of released exosomes per HEK-Blue hTLR3 cell increased with the concentration of poly(I:C), but the number was unchanged in T-REx Mock and T-REx AspALP cells (Figure 4B).

### 2.6. Confirmation That the ER Fraction Is Not Contaminated in Exosomes

To exclude the possibility that the ER fraction was contaminated by the exosome fraction, we examined whether exosomes contained ER-associated proteins, such as Grp78 and ATF6, by Western blotting (Figure 5A). Exosome-associated proteins, such as CD81 and Alix, were detected in exosomes, but Grp78 and ATF6 were not. In addition, we examined SEAP SPs (full length and N-terminal fragment) in the ER fractions. Peak sizes at *m*/*z* 1878 and 1277 were not found in the ER fraction from HEK-Blue hTLR3 cells in the presence or absence of poly(I:C). These results indicated that the N-terminal SP fragment was contained in exosomes from HEK-Blue hTLR3 cells after poly(I:C) treatment and those exosomes did not contaminate the ER fraction.

### 2.7. Binding of SP Fragment with CALM

In a previous report, preprolactin SP was shown to bind to CALM, a multifunctional calcium-binding messenger protein [17]. To clarify whether the exosomal SP fragment could bind to CALM, we examined the binding of hAPP SP (9–17), a C-terminal fragment of APP SP, with CALM (Figure 6). Peak sizes at *m*/*z* 972, indicating hAPP SP (9–17), and *m*/*z* 861 were found. In addition, the peak size at *m*/*z* 861 was significantly increased in the presence of Ca^2+^ (Figure 6A,B). MS/MS analysis identified the peak at *m*/*z* 859 +/− 4 as the C-terminal fragment of APP SP (LAAWTARA), which contains a -TAR- amino acid sequence (Figure 6C). Furthermore, in the presence of CALMIP, the peak size at *m*/*z* 861 was significantly decreased and the peak size at *m*/*z* 2073, corresponding to CALMIP, was significantly increased (Figure 6D). These results indicate that SP fragments in exosomes can bind to CALM (Figure 6E).

## 3. Discussion

In this paper, we demonstrated that the N-terminal fragment of SEAP SP is included in exosomes and secreted into the extracellular fluid of HEK-Blue hTLR3 cells after poly(I:C) stimulation. As the poly(I:C) concentration was increased, the amount of SP fragments in the exosomes increased (Figure 2). Since the synthesis of SEAP and its SP are increased when HEK-Blue hTLR3 cells are stimulated with poly(I:C) (Figure 1B), the amount of SEAP SP fragment in exosomes is probably increased too. Exosomes encapsulate noncoding RNAs such as micro RNAs, long non-coding RNAs, and circular RNAs and functional proteins and play an important role in intercellular signaling in pathological conditions as well as physiological states [18,19,20,21,22]. Since SPs and their fragments have been shown to have bioactivity [5,6,8,9], it is possible that SP fragments in exosomes are also involved in intercellular signaling.

SP-containing synthesizing proteins can insert into the ER membrane in either type I or type II orientation and are cut by signal peptidases to remove the signal sequence [12] on the membrane. Then, SPs are cut by SPP to produce two fragments [13]. SPs of type I membrane proteins penetrate the N-terminus across the ER membrane while the C-terminus remains in the cytoplasm. By contrast, SPs of type II membrane proteins transfer the C-terminus of the protein across the membrane, while the N-terminus remains on the cytoplasmic side. Determination of the orientation (type I or type II) is dependent on the features of the SP sequences [23,24]. We previously reported that the C-terminal but not the N-terminal fragment of type I-oriented APP SP is found in exosomes and secreted into the extracellular fluid of Dox-treated T-Rex AspALP cells [16]. On the other hand, we found the N-terminal but not the C-terminal fragment of type II-oriented SEAP SP in the exosomes of HEK-Blue hTLR3 cells (Figure 2). These findings indicate that cleaved SP fragments located at the cytoplasmic side are transferred to exosomes.

As shown in the ER fractions of T-REx AspALP cells in a previous study [16], neither the N-terminal nor the C-terminal of SP fragments are detected in the ER fractions of HEK-Blue hTLR3 cells according to MALDI-TOF MS analysis (Figure 5), suggesting that cleaved SP fragments are degraded in the ER or rapidly moved out of it.

It remains unclear how some SP fragments are included in the exosomes from the ER. Exosomes are membrane vesicles released to the extracellular fluid upon the exocytic fusion of multivesicular endosomes with the cell surface. They have a particular composition reflecting their origin in endosomes as intraluminal vesicles [18]. Therefore, it is likely that SP fragments have some mechanism by which they are transported from the ER to endosomes without being degraded. In a cell-free system, SP fragments of preprolactin interact with CALM [17]. In this study, we found that C-terminal fragments of APP SP bound to CALM and that this binding was increased in the presence of Ca^2+^ (Figure 6). CALM is a ubiquitous calcium-binding protein that regulates many types of proteins and is involved in a variety of cellular processes such as immune responses, metabolism, higher brain function, apoptosis and intracellular migration [25]. In addition, there is a CALM-dependent translocation pathway for small secretory proteins via the binding of SPs [26]. Some groups reported that exosomes isolated from various cells contained CALM [27,28]. Moreover, we found C-terminal fragments of APP SP bound to CALM in T-REx AspALP cells and their exosomes (manuscript in preparation). Therefore, adaptor molecules, such as CALM, may interact with SP fragments to facilitate their transport to endosomes and/or exosomes (Figure 6D).

When T-REx Mock, T-REx AspALP, and HEK-Blue hTLR3 cells were stimulated with poly(I:C), the number of cells slightly decreased. These cell lines are based on HEK293 cells that express TLR3 [29]. Since poly(I:C), a TLR3 ligand, inhibits cell growth [30], the decrease in cell number could be due to the effect of endogenous TLR3.

Although the intracellular signaling system in HEK-Blue hTLR3 cells is artificially constructed based on the NFκB signaling pathway, TLR signal generation systems and the NFκB signaling pathway are common in many cell types. When HEK-Blue hTLR3 cells were treated with poly(I:C), the N-terminal fragment of SEAP SP in exosomes was increased in a concentration-dependent manner, suggesting that at least some SP fragments are transported to exosomes by intracellular signaling and are exosomal components.

Some SPs and SP fragments in body fluids are potential biomarkers for a variety of diseases [1,2,31]. In many cases, they are detected from the blood or serum. If they are released by the same mechanism as in this study, then isolating exosomes may enable more sensitive biomarker detection.

## 4. Materials and Methods

### 4.1. Cells

T-REx Mock, T-REx AspALP and HEK-Blue hTLR3 cells were used in this study. T-REx Mock and T-REx AspALP cells are HEK-293 cells expressing tetracycline (Tet) repressor and made using the T-REx System (Thermo Fisher Scientific, Waltham, MA, USA) [32]. The APP SP-SEAP sequence, in which the SP sequence of SEAP was replaced with the SP sequence of APP, was inserted into multicloning sites of pcDNA 4/TO vector, and T-REx AspALP cells, in which the modified vector was genetically transferred, were picked up as a stable mutant strain. T-REx Mock cells, in which unmodified pcDNA 4/TO vector was genetically transferred, were generated as a stable strain. HEK-Blue hTLR3 cells (InvivoGen, San Diego, CA, USA) are HEK293 cells co-transfected with human TLR3 genes and inducible SEAP reporter genes. HEK-Blue hTLR3 cells (1 × 10^6^ in total) were plated on a 100-mm Falcon cell culture dish (Corning Inc., Corning, NY, USA) and cultured in Dulbecco’s Modified Eagle’s Medium (DMEM) (Sigma-Aldrich, St Louis, MO, USA) supplemented with 10% fetal bovine serum, from which extracellular vesicles were removed by centrifugation at 110,000× *g* for 24 h, and Penicillin-Streptomycin (Thermo Fisher Scientific) in the presence or absence of 0–10 μg/mL polyinosinic-polycytidylic acid (poly(I:C)) (Bio-Techne, Minneapolis, MN, USA) for 72 h at 37 °C in 5% CO_2_/95% humidified air. T-REx Mock and T-REx AspALP cells (1 × 10^6^ in total) were also plated on dishes and cultured in modified medium, into which was added 1 μg/mL doxycycline (Dox) (Takara Bio Inc., Kusatsu Shiga, Japan) for HEK-Blue hTLR3 cells in the presence or absence of 10 μg/mL poly(I:C) for 72 h at 37 °C in 5% CO_2_/95% humidified air. Photographs of the cells were taken using a microscope equipped with a DP73 camera (Olympus, Tokyo, Japan). Each conditioned medium and cells were collected from one dish.

### 4.2. Isolation of Exosomes from Conditioned Medium

Conditioned medium was centrifuged at 300× *g* for 5 min at 4 °C to remove live cells, and the supernatant was centrifuged at 2000× *g* for 20 min to remove apoptotic vesicles. Microvesicles were removed by centrifugation at 10,000× *g* for 60 min. Exosomes were prepared from the supernatant by centrifugation at 110,000× *g* for 70 min and re-suspended in 20 μL of PBS. The cells were washed in PBS twice after cell counting and stocked as cell pellets.

### 4.3. Measurement of SEAP Activity

SEAP activity in conditioned medium was measured using QUANTI-Blue Solution (InvivoGen). In brief, 20 μL of conditioned medium was mixed with 180 μL of QUANTI-Blue Solution and incubated at 37 °C for 2 h in a 96-well plate. Absorbance at 620 nm was measured using a microplate reader.

### 4.4. Nanoparticle Tracking Analysis (NTA)

The number and average size of exosomes were measured using a NanoSight NS300 (Malvern Panalytical Ltd., Malvern, UK). Exosomes were diluted at 1:100 in degassed water to a final volume of 600 μL and applied through a syringe for measurement. The camera level was increased until all particles were distinctly visible without exceeding a particle signal saturation over 20% (levels 14–16). Automatic settings for the maximum jump distance and blur settings were utilized. The detection threshold was 5. For each measurement, five 60-s videos were captured under the following conditions: cell temperature, 25 °C; syringe pump speed, 100 (instrument-specific unit); camera, sCMOS; laser, 488 nm blue. After capturing, the number and size of exosomes were analyzed using NanoSight NTA 3.2 software build 3.2.16. Released exosomes (particles per cell) were calculated using data from the NTA and cell counting.

### 4.5. Isolation of ER Fraction

The ER fraction was isolated from pellets of HEK-Blue hTLR3 cells (total 3 × 10^7^ cells) using the Endoplasmic Reticulum Enrichment Extraction Kit (Novus Biologicals, Centennial, CO, USA). In brief, the cell pellets were mixed with 1 mL of 1× Isosmotic Homogenization Buffer followed by 10 μL of 100× Protease Inhibitor Cocktail. The cells were then homogenized by moving them in and out of a pipette 100 times. The homogenate was centrifuged at 1000× *g* for 10 min at 4 °C, and the supernatant was centrifuged at 12,000× *g* for 15 min at 4 °C. The supernatant was then centrifuged at 90,000× *g* for 70 min at 4 °C and discarded. The pellet was suspended in 200 μL of 1× Suspension Buffer supplemented with 2 μL of 100× Protease Inhibitor Cocktail as the total ER fraction.

### 4.6. SP Identification with MALDI-TOF MS/MS

Exosomes and the ER fraction (10 μL each) were dissolved in 90 μL of 8 M urea. Peptides were concentrated from the solution with GL-Tip SDB and GC columns (GL Sciences Inc., Tokyo, Japan) and eluted with 20 μL of 80% acetonitrile supplemented with 0.1% trifluoroacetate. The peptide solution was mixed at 1:1 with 10 mg/mL CHCA in 50% acetonitrile supplemented with 0.1% trifluoroacetate, and 1 μL of the mixture was plated on an MTP 384 target plate ground steel (Bruker, Billerica, MA, USA). After the plate was air-dried, the peptides were measured using an ultrafleXtreme MALDI-TOF MS (Bruker) and analyzed using Flex analysis software (Bruker). In the MS/MS analysis, amino acid sequences were determined within an error of 0.7 Da.

### 4.7. Western Blotting

The cells, exosomes, and ER from HEK-Blue hTLR3 cells were lysed in RIPA buffer (20 mM Tris-HCl, 150 mM NaCl, 1 mM Na_2_EDTA, 1 mM EGTA, 1% NP-40, 1% sodium deoxycholate, 2.5 mM sodium pyrophosphate, 1 mM β-glycerophosphate, 1 mM Na_3_VO_4_, and 1 μg/mL leupeptin, pH 7.5) (Cell Signaling Technology, Danvers, MA, USA) by sonication in iced water. The BCA Protein Assay (Thermo Fisher Scientific, Rockford, IL, USA) was performed to determine protein concentrations. The cells were mixed with loading buffer with or without dithiothreitol (Cell Signaling Technology) to 2 mg/mL, exosomes to 0.2 mg/mL, and ER to 0.5 mg/mL. The mixture was boiled for 5 min, and 5 μL was separated by SDS-PAGE on mini-gels (Oriental Instruments, Kanagawa, Japan). The separated proteins were then transferred to polyvinylidene difluoride membranes on an iBlot2 Gel Transfer Device (Thermo Fisher Scientific) and blocked in TBS (Tris-buffered saline) supplemented with 5% nonfat dry milk (Cell Signaling Technology) and 0.1% Tween 20 for 1 h at room temperature. Membranes were then incubated in the presence of primary antibodies diluted in Can Get Signal Immunoreaction Enhancer Solution 1 (Toyobo, Osaka, Japan) at a 1:6000 dilution overnight at 4 °C. Primary antibodies directed against CD81 (NB100-65805, Novus Biologicals, Centennial, CO, USA) for nonreducing conditions and against Alix (#92880, Cell Signaling Technology), Grp78 (GTX113340, GeneTex, Irvine, CA, USA) and ATF6 (24169-1-AP, Proteintech, Rosemont, IL, USA) for reducing conditions were used. Then, the membranes were washed 4 times for 8 min with TBS-T (TBS supplemented with 0.1% Tween 20) and incubated in Can Get Signal Immunoreaction Enhancer Solution 2 (Toyobo) containing HRP-conjugated secondary antibodies (anti-mouse IgG (#7076) and anti-rabbit IgG (#7074), Cell Signaling Technology) at a 1:12,000 dilution for 1 h at room temperature. The proteins were visualized by chemiluminescence using Clarity Western ECL Substrate (Bio-Rad, Hercules, CA, USA) and Light Capture II (Atto, Tokyo, Japan).

### 4.8. Calmodulin Binding Assay

hAPP SP (9–17) (LLAAWTARA) was purchased from Eurofins Scientific (Luxembourg). hAPP SP (9–17) (500 nM) was gently mixed with recombinant human calmodulin (CALM) 1/2/3 (1 μg/mL) (Abcam, Cambridge, UK), which contained a histidine tag at N-terminus, in xTractor buffer (200 μL) (Takara) in the presence or absence of CaCl_2_ (500 μM) and calmodulin inhibitory peptide (CALMIP, Sigma-Aldrich, 2.5 μM) overnight at 4 °C. CALM solution (150 μL) was separated from the mixture using a Capturem His-Tagged Purification Miniprep (Takara) following the manufacturer’s instructions. CALM solution (100 μL ) was dissolved in 100 μL of 8 M urea. Peptides were concentrated from the solution with GL-Tip SDB and GC columns and eluted with 20 μL of 80% acetonitrile supplemented with 0.1% trifluoroacetate. Eluted SPs were analyzed by MALDI-TOF MS/MS.

### 4.9. Statistical Analysis

Statistical analyses were performed using a two-tailed *t*-test or one-way ANOVA and Tukey post-hoc tests. Differences were considered significant when the *p*-value was less than 0.05.

## 5. Conclusions

SP fragments can be secreted out of cells via exosomes. Depending on the SP sequence, the exosomes contain the N-terminal or C-terminal of SP fragments. Since exosomes play an important role in intercellular signal transduction, SP fragments in exosomes may have significant physiological functions in the recipient cells.

## Figures and Tables

**Figure 1 ijms-23-03137-f001:**
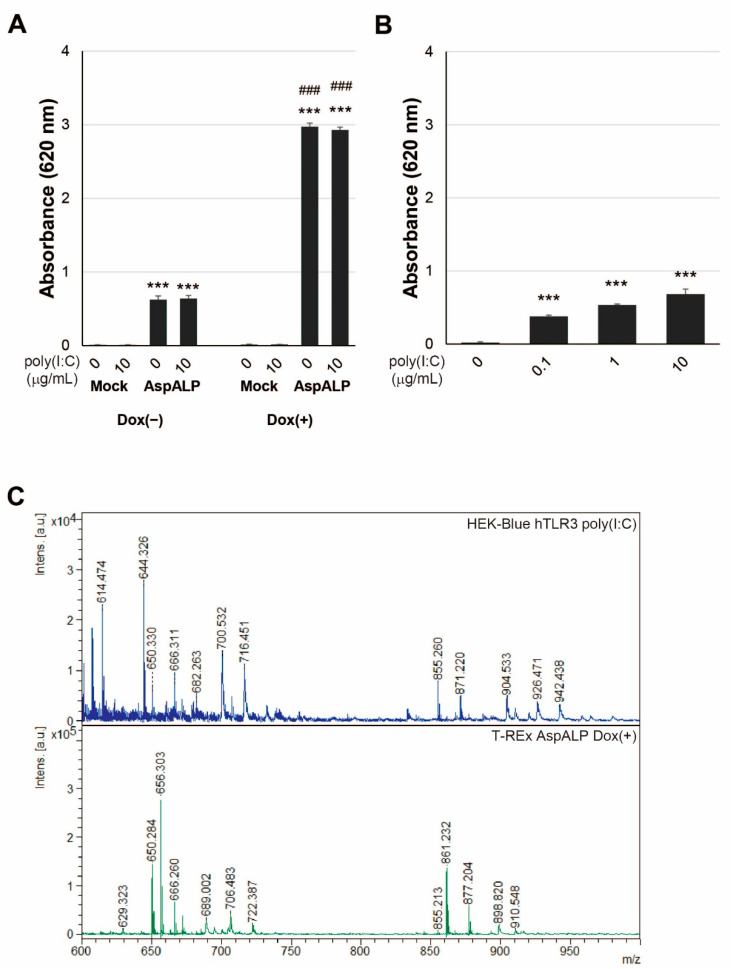
SEAP is released from poly(I:C)-stimulated cells. (**A**) The SEAP activity of T-REx Mock (Mock) and T-REx AspALP (AspALP) cells in the presence or absence of Dox at 1 μg/mL or poly(I:C) at 10 μg/mL. *** *p* < 0.001 vs. T-REx Mock cells in the absence of poly(I:C). ^###^ *p* < 0.001 vs. T-REx AspALP cells in the absence of Dox. (**B**) The SEAP activity of HEK-Blue hTLR3 cells in the presence of poly(I:C) at 0–10 μg/mL. *** *p* < 0.001 vs. HEK-Blue hTLR3 cells in the absence of poly(I:C). (**C**) The peptide solution extracted from exosomes of HEK-Blue hTLR3 cells 3 days after poly(I:C) treatment at 10 μg/mL and T-REx AspALP cells 3 days after Dox treatment at 1 μg/mL was analyzed by MALDI-TOF MS.

**Figure 2 ijms-23-03137-f002:**
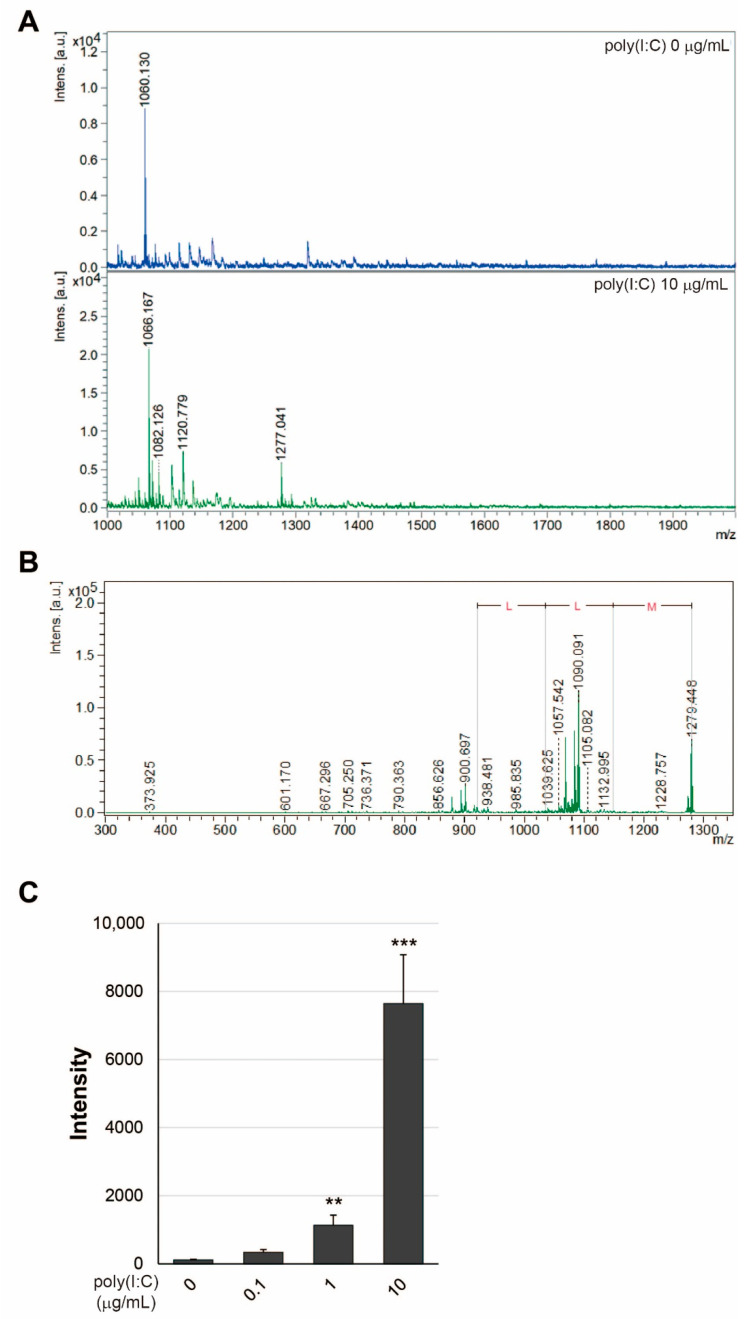
N-terminal fragments of SEAP SP in exosomes released from HEK-Blue hTLR3 cells. (**A**) The peptide solution extracted from exosomes of HEK-Blue hTLR3 cells in the presence of poly(I:C) at 0–10 μg/mL for three days was analyzed by MALDI-TOF MS. (**B**) MS/MS analysis of peaks at *m*/*z* 1277 +/− 6 was performed. (**C**) The peak intensity at 1277 was measured by MALDI-TOF MS. ** *p* < 0.01, *** *p* < 0.001 vs. the absence of poly(I:C).

**Figure 3 ijms-23-03137-f003:**
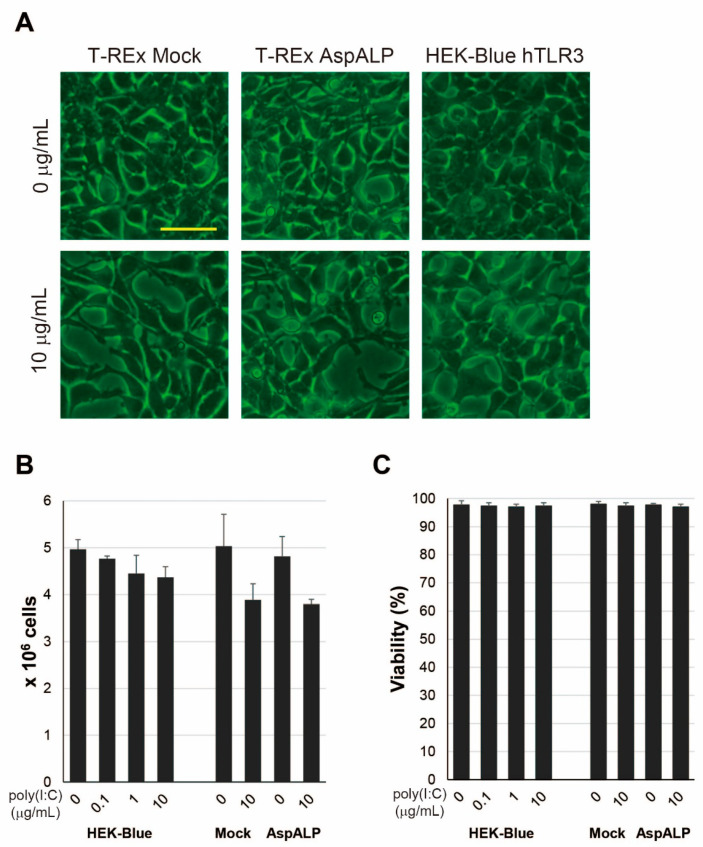
Morphology and growth in cells stimulated with poly(I:C). (**A**) Morphology of T-REx Mock (Mock), T-REx AspALP (AspALP), and HEK-Blue hTLR3 (HEK-Blue) cells after three days in the presence or absence of poly(I:C) at 10 μg/mL. Scale bar indicates 50 μm. The number (**B**) and viability (**C**) of cells after three days in the presence or absence of poly(I:C) at 0–10 μg/mL.

**Figure 4 ijms-23-03137-f004:**
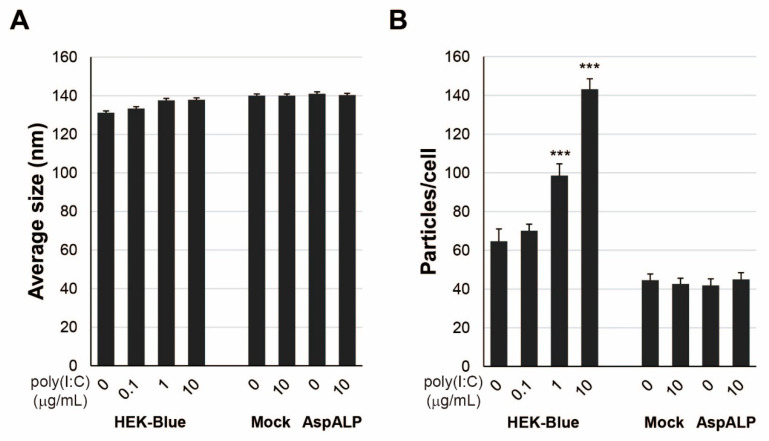
Properties of exosomes released from poly(I:C)-stimulated cells. (**A**) The average size of exosomes released from poly(I:C)-stimulated HEK-Blue hTLR3, T-REx Mock and T-REx AspALP cells. (**B**) Released exosomes per cell. *** *p* < 0.001 vs. cells in the absence of poly(I:C).

**Figure 5 ijms-23-03137-f005:**
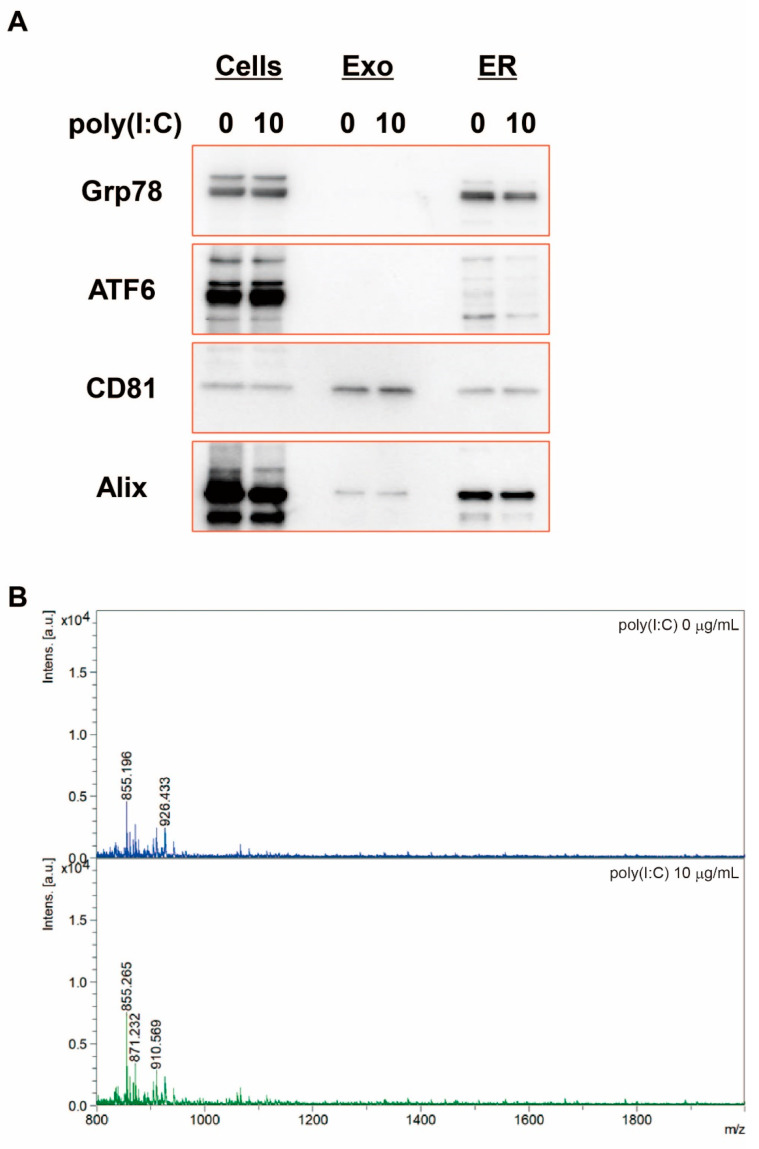
Exosomes do not contaminate the ER fraction from HEK-Blue hTLR3 cells. (**A**) Western blot analysis of markers for the ER and exosomes (Exo) fractions from HEK-Blue hTLR3 cells in the presence or absence of poly(I:C). (**B**) The ER fraction from HEK-Blue hTLR3 cells after poly(I:C) treatment was analyzed by MALDI-TOF MS.

**Figure 6 ijms-23-03137-f006:**
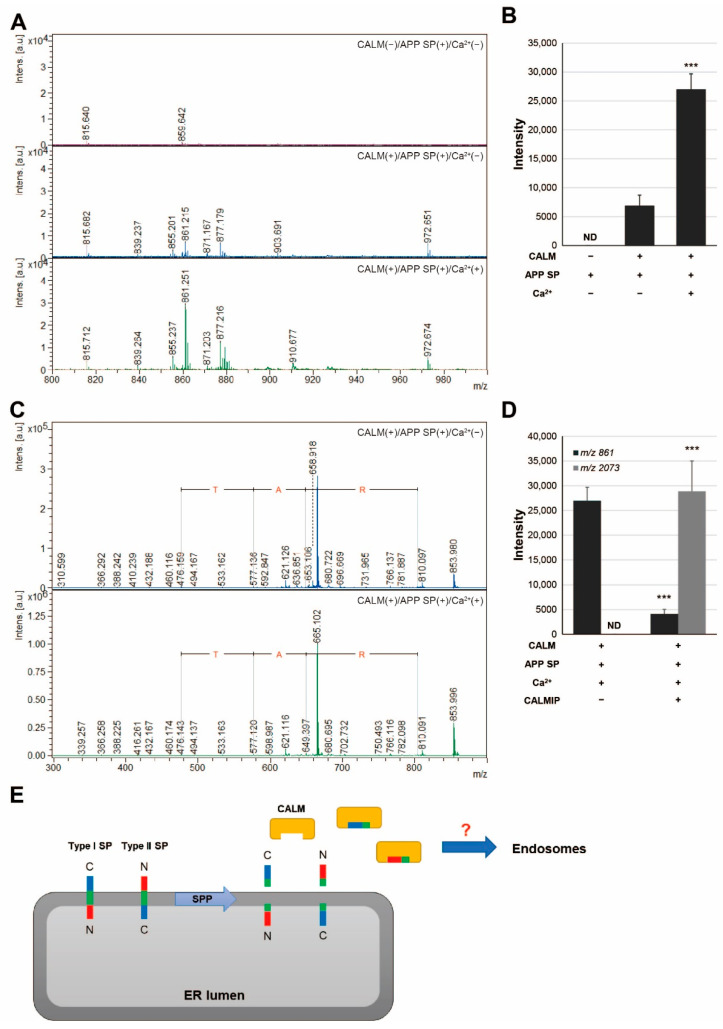
Binding of exosomal SP fragments with CALM in the presence of Ca^2+^. (**A**) The peptide solution extracted from His-tagged CALM was analyzed by MALDI-TOF MS. (**B**) The peak intensity at 861 was measured by MALDI-TOF MS. ND, not detected. *** *p* < 0.001 vs. CALM(+)/APP SP(+)/Ca^2+^(−). (**C**) MS/MS analysis of the peaks at *m*/*z* 859 +/− 4. (**D**) The peak intensity at 861 and 2073 in the presence or absence of CALMIP was measured by MALDI-TOF MS. ND, not detected. *** *p* < 0.001 vs. CALM(+)/APP SP(+)/Ca^2+^(+)/CALMIP(−); (**E**) the working hypothesis of this study.

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
