# Peer review of "Signal Sequence-Dependent Orientation of Signal Peptide Fragments to Exosomes"

_ijms, 2022, doi:10.3390/ijms23063137_

Round 1
Reviewer 1 Report
In this paper Kenji Ono et al have used a reporter HEK-Blue hTLR3 cells that expresses both human toll-like receptor 3 gene and inducible a type-II membrane protein, secreted alkaline phosphatase (SEAP) gene and described that an N-terminal fragment of the signal peptide from SEAP, is present in exosomes and then secreted into the extracellular fluid. The result of this study are interesting but the experiments are mostly in line with the previous published reports on type-I membrane protein Ono K et al doi:10.1016/j.bbrc.2021.04.073. To warrant publication in IJMS, the authors need to illustrate more on the mechanism and on the components of cellular machinery involved in the regulation of this pathway.
- To ascertain the involvement of calmodulin in the transportation of signal peptide fragments from the endoplasmic reticulum to exosomes, the authors could perform their experiments upon chemical/genetic (knockdown) inhibition of calmodulin to find out what happens to the transport of signal peptide fragments in the absence of calmodulin. The authors could also demonstrate this for both the signal peptide from type-1 membrane protein (APP) and from type-II membrane protein (SEAP)?
- Is this study applicable to all signal peptides from type-II/type-I membrane protein (i.e. an N –term fragment of the signal peptide from a type-II membrane protein is always contained in exosomes while a C-term fragment of the signal peptide is contained in exosomes from a type-I membrane protein)? The authors could test this using different reporter in their inducible cell line system.
- In Figure 6, authors have shown the binding of the C-term fragment of APP SP to Calmodulin by MALDI/TOF MS. The authors could also test this experiment to analyze the binding of the N-term fragment of SEAP SP to calmodulin (as depicted in their model fig 6D).
Author Response
Response to Reviewer #1
Thank you very much for reviewing our manuscript and offering constructive insights.
We have addressed your comments with point-by-point responses, and revised the manuscript accordingly.
- To ascertain the involvement of calmodulin in the transportation of signal peptide fragments from the endoplasmic reticulum to exosomes, the authors could perform their experiments upon chemical/genetic (knockdown) inhibition of calmodulin to find out what happens to the transport of signal peptide fragments in the absence of calmodulin. The authors could also demonstrate this for both the signal peptide from type-1 membrane protein (APP)and from type-II membrane protein (SEAP)?
Response: We appreciate the reviewer’s comment on this point. We agree that additional information on chemical/genetic (knockdown) inhibition of calmodulin as the reviewer suggested would be valuable. Regrettably, we cannot show the additional information in this manuscript. Although we have been trying to inhibit calmodulin using siRNA and inhibitors such as calmidazolium chloride, the absence of calmodulin induces cell death and we cannot find good experimental conditions because calmodulin plays important roles in multiple intracellular functions as well as binding to signal peptides. Accordingly, we show binding of calmodulin to signal peptides on cell-free system in this manuscript. To further clarify binding of calmodulin to signal peptides, we added results with calmodulin inhibitory peptides (as shown in Fig. 6D).
- Is this study applicable to all signal peptides from type-II/type-I membrane protein (i.e. an N –term fragment of the signal peptide from a type-II membrane protein is always contained in exosomes while a C-term fragment of the signal peptide is contained in exosomes from a type-I membrane protein)? The authors could test this using different reporter in their inducible cell line system.
Response: Thank you for your good suggestion. We agree one of important questions, however it is difficult to give an answer whether this study is applicable to all signal peptides clearly. To clarify whether signal peptides from type-II membrane protein as well as type-I membrane protein are contained in exosomes, we used SEAP-reporter system. On the other hand, we have some evidence that signal peptides from endogenous proteins are detectable in exosomes from some cells without genetical transfection. We believe that secretion of signal peptides via exosomes is one of physiological functions inherently possessed by cells, however it is not clear whether all signal peptides are moved to exosomes or secretion of signal peptides via exosomes is induced dependent on cellular activation. This point needs to be clarified in future ongoing studies. Therefore, we wrote “suggesting that at least some SP fragments are transported to exosomes by intracellular signaling and are exosomal components.” (p. 8 line 310-311)
- In Figure 6, authors have shown the binding of the C-term fragment of APP SP to Calmodulin by MALDI/TOF MS. The authors could also test this experiment to analyze the binding of the N-term fragment of SEAP SP to calmodulin (as depicted in their model fig 6D).
Response: We thank the reviewer for this comment. In this manuscript, we want to show that signal peptides from type-II membrane protein as well as type-I membrane protein are contained in exosomes as a novel aspect. In addition, we want to show a possibility that adaptor molecules such as calmodulin are concerned in transport to endosomes. Therefore, we showed Fig. 6E as the working hypothesis of this study. We will show that binding of N-terminal fragment of type-II signal peptides to adaptor molecules such as calmodulin in future study.

Reviewer 2 Report
Ono and colleagues presented an interesting research article aimed at evaluating the signal peptide fragments to exosomes. For this purpose, the authors performed different functional experiments on cell lines demonstrating that an N-terminal fragment of the SP of SEAP protein is contained in exosomes and secreted into the extracellular fluid. Overall, the manuscript is very interesting and the experimental design is well-conceived. Below are reported some comments that will improve the quality of the manuscript:
1) In the first sentence of the Introduction section, the authors state: “Recent studies have demonstrated the novel potential roles of signal peptides (SPs) and their fragments as biomarkers and substances with physiological activity.”. Please indicate which studies and add appropriate references;
2) In the Chapter “Western Blot” of the Methods section, please indicate the catalog number of the antibodies used and each dilution;
3) Please add Figures within the text when you mention that specific result;
4) Please briefly mention other molecules encapsulated within exosomes, like lncRNA, circRNA, etc., which play a fundamental role in signaling, especially in human diseases. For this purpose, please see:
- PMID: 34198978
- PMID: 33488093
- PMID: 30643155
- PMID: 32269609
Author Response
Responses to Reviewer #2
Thank you very much for reviewing our manuscript and offering constructive insights.
We have addressed your comments with point-by-point responses, and revised the manuscript accordingly.
- In the first sentence of the Introduction section, the authors state: “Recent studies have demonstrated the novel potential roles of signal peptides (SPs) and their fragments as biomarkers and substances with physiological activity.”. Please indicate which studies and add appropriate references;
Response: Thank you for your suggestion. We inserted appropriate references (p.1 line 35).
- In the Chapter “Western Blot” of the Methods section, please indicate the catalog number of the antibodies used and each dilution;
Response: Thank you for your suggestion. We inserted the catalog number of the antibodies in the Chapter ”Western Blot” of the Methods section (p. 4 line 157-163).
- Please add Figures within the text when you mention that specific result;
Response: Thank you for your suggestion. We inserted Figures within the text (p. 7 line 260, 261and 279, p.8 line283 and 292).
- Please briefly mention other molecules encapsulated within exosomes, like lncRNA, circRNA, etc., which play a fundamental role in signaling, especially in human diseases. For this purpose, please see: PMID34198978, PMID33488093, PMID30643155, PMID32269609
Response: Thank you for your suggestion. We modified sentences and added references (p. 7 line 262-265).

Round 2
Reviewer 1 Report
Accept
Reviewer 2 Report
The authors well-addressed all my previous comments. The manuscript was significantly improved thus it can be accepted for publication after the editorial check.